# The Role of Allogeneic Hematopoietic Stem Cell Transplantation in Pediatric Leukemia

**DOI:** 10.3390/jcm10173790

**Published:** 2021-08-25

**Authors:** Mattia Algeri, Pietro Merli, Franco Locatelli, Daria Pagliara

**Affiliations:** 1Department of Pediatric Hematology and Oncology, Scientific Institute for Research and Healthcare (IRCCS), Bambino Gesù Childrens’ Hospital, 00165 Rome, Italy; pietro.merli@opbg.net (P.M.); franco.locatelli@opbg.net (F.L.); daria.pagliara@opbg.net (D.P.); 2Department of Pediatrics, Sapienza University of Rome, 00185 Rome, Italy

**Keywords:** allogeneic stem cell transplantation, acute lymphoblastic leukemia, acute myeloid leukemia, minimal residual disease, conditioning regimen, alternative donors

## Abstract

Allogeneic hematopoietic stem cell transplantation (HSCT) offers potentially curative treatment for many children with high-risk or relapsed acute leukemia (AL), thanks to the combination of intense preparative radio/chemotherapy and the graft-versus-leukemia (GvL) effect. Over the years, progress in high-resolution donor typing, choice of conditioning regimen, graft-versus-host disease (GvHD) prophylaxis and supportive care measures have continuously improved overall transplant outcome, and recent successes using alternative donors have extended the potential application of allotransplantation to most patients. In addition, the importance of minimal residual disease (MRD) before and after transplantation is being increasingly clarified and MRD-directed interventions may be employed to further ameliorate leukemia-free survival after allogeneic HSCT. These advances have occurred in parallel with continuous refinements in chemotherapy protocols and the development of targeted therapies, which may redefine the indications for HSCT in the coming years. This review discusses the role of HSCT in childhood AL by analysing transplant indications in both acute lymphoblastic and acute myeloid leukemia, together with current and most promising strategies to further improve transplant outcome, including optimization of conditioning regimen and MRD-directed interventions.

## 1. Introduction

Despite the remarkable achievements obtained with frontline chemotherapy in the treatment of children with acute leukemia (AL), a significant proportion of patients still are treated with allogeneic hematopoietic stem cell transplantation (HSCT), either in first complete remission (CR1) or beyond, to achieve definitive disease eradication [1,2,3,4,5]. Over the years, the outcome of allogeneic HSCT for AL has continuously improved thanks to progress in high-resolution HLA typing, choice of conditioning regimen and supportive care measures. Furthermore, mainly owing to fundamental advances in graft manipulation techniques and graft-versus-host disease (GvHD) prophylaxis, results obtained using alternative donors are no longer inferior to those achieved using HLA-identical siblings and fully matched volunteers [6]. At the same time, successes in the transplant field are constantly paralleled by refinements in chemotherapy protocols, which take advantage of continuous breakthroughs of genomic medicine to achieve better treatment stratification and identify genetic lesions targetable with precision medicine approaches [7,8]. Such targeted approaches may be employed, in their turn, to further improve HSCT outcome by inducing a better remission status before transplant, especially in light of the robust evidence showing that pre- and post-HSCT minimal residual disease (MRD) represents one of the major determinants of subsequent relapse and long-term prognosis in AL [9,10]. This observation is particularly relevant in B-cell acute lymphoblastic leukemia (ALL), owing to the availability of highly effective immunotherapy approaches capable of inducing MRD-negative remissions of otherwise refractory and untreatable diseases [11]. The purpose of this review is to discuss the current role of HSCT in childhood AL by scrutinizing transplant indications in both ALL and acute myeloid leukemia (AML). We also discuss most promising practices to further improve transplant outcome, by analyzing (1) the predictive role of MRD and potential MRD-directed interventions, before and after HSCT; (2) the choice of conditioning regimen and (3) most recent results obtained with the use alternative donors.

## 2. HSCT Indications in ALL

Since most children with ALL have a good prognosis with current frontline protocols, only a minority of these patients are eligible for HSCT in CR1. On the contrary, most patients who experience a relapse are candidates for HSCT [12]. Despite the criteria used to assess this indication may vary between different cooperative groups, most studies/protocols consider eligible children with an estimated EFS probability lower than 50%, according to response to induction treatment and specific biologic features. Advances in biological characterization, improved MRD measurement capacity and its incorporation in treatment protocols to optimize risk stratification, and the upfront introduction of new biological agents (e.g., blinatumomab in AIEOP-BFM (EUDRACT 2016-001935-12) and COG protocols (AALL1731) and inotuzumab (AALL1732) and tisagenlecleucel (AALL1721/CASSIOPEIA) in COG protocol) have changed (and will continue to shape) the indications for HSCT over time [13]. Current HSCT indications in childhood ALL are summarized in Table 1. For example, criteria that were considered in past studies, such as a high white blood cell (WBC) count at presentation, Philadelphia chromosome positivity (Ph+), or poor-prednisone response, are no longer considered strict indications for transplantation [13]. The strongest criterion for HSCT indication is the response of the disease to induction therapy, this being a surrogate marker of leukemia sensitivity to chemotherapy. Indeed, MRD at selected time points is currently the single most powerful prognostic factor in childhood ALL [14]. Both multiparametric flow cytometry or quantitative PCR for specific rearrangements of immune-receptor genes IgH/TCR are currently used for MRD assessment [15]. The role of MRD for identification of HSCT candidates has been clearly proven by several groups, including The Italian Association of Pediatric Haematology–Oncology (AIEOP)/Berlin–Frankfurt–Muenster (BFM) group [16].

Additional criteria are employed to identify those patients who would benefit from HSCT in the first CR. Differing from AML, PIF is infrequent in pediatric ALL (being observed in less than 2% of patients); regardless, its occurrence represents a strong indication for HSCT in CR1, as shown by Schrappe and colleagues in a large retrospective study [17]. Hypodiploidy (<44 chromosomes) defines a high-risk group of patients with a poor outcome (the lowest the number of chromosome, the worst the prognosis) [18]. Thus, hypodiplody has been considered a strong indication to HSCT, although two recent retrospective studies, one from COG [19], the other from 16 cooperative groups [20] did not show an advantage for patients transplanted in CR1. Patients carrying the t(17;19)(q22;p13) translocation, resulting into the fusion gene TCF3-HLF (E2A-HLF), have a dismal outcome, irrespectively of MRD clearance [21]. For this reason, these patients are considered eligible for experimental therapies (e.g., BCL-2 inhibitors or anti-CD19-directed immunotherapies) and for HSCT in CR1 [22]. Deletions of Ikaros (IKZF1), a zinc-finger transcription factor required for the development of all lymphoid lineages, have been shown to confer an increased risk of recurrence [23]. In a recent paper, Stanulla and colleagues showed that IKZF1 deletions that co-occurred with deletions in CDKN2A, CDKN2B, PAX5 or PAR1 in the absence of ERG deletion were associated with the worst outcome and were grouped as IKZF1^plus^ [24]. The IKZF1^plus^ prognostic effect differed dramatically according to the MRD levels after induction treatment; in particular, IKZF1^plus^ patients with intermediate and high-risk MRD had a miserable outcome and are now considered candidates to receive HSCT in CR1. Rearrangements involving the KMT2A (MLL) gene on chromosome 11q23 are observed in a large part of infants ALL; on the contrary, in children older than 1 year, they are much less frequent [25]. KMT2A-rearrangments represent a high-risk feature [26] justifying the use of HSCT in CR1, although for children with these abnormalities the indication to HSCT is determined in combination with (i) age less than 6 months and either poor response to steroids or leukocytes ≥300 × 10^9^/L at diagnosis [27] for infant ALL, and (ii) MRD clearance. Intrachromosomal amplification of chromosome 21 is a rare recurrent lesion (found in less than 3% cases of pediatric ALL). The sole use of standard chemotherapy in this population has been associated with an increased risk of relapse [28]. Studies conducted in the UK have reported improved survival with the use of front-line HSCT [29]. Since patients with T-ALL cannot still benefit from immunotherapy approaches (including CAR-T cells and BiTEs), they have a lower chance of being rescued when a relapse occurs. In addition, for this reason, patients with poor MRD clearance at the end of induction are considered candidates for HSCT in CR1.

Indications to HSCT for children with ALL who experience a first relapse are currently based on disease immunophenotype (T/B ALL) time elapsing between diagnosis and recurrence and site of relapse [30,31,32,33]. Since persistence of MRD after induction/consolidation therapy has been proven to influence outcome in relapsed children [33,34,35,36], MRD measurement has been incorporated in the decision process for treatment stratification and indication to HSCT. In addition, patients with TP53 mutations, t(1;19), t(17;19), hypodiploidy and KMT2A-rearragments have a dismal prognosis regardless of the time elapsing between diagnosis and relapse and should be offered HSCT once a new CR is achieved. Overall, two-thirds of children in CR2 proceed to HSCT. Finally, all patients who have experienced two or more relapses are considered candidates to receive a HSCT, regardless of the type of donor available.

## 3. HSCT Indications in AML

The use of HSCT as a consolidation strategy for pediatric patients with AML in CR1 has been the subject of much debate [37,38,39]. Historically, HSCT in CR1 was reserved for those children who had an available, HLA-identical sibling donor. Early studies adopting this Mendelian/genetic randomization strategy reported a reduction in leukemia relapse-rate (RR) in patients receiving transplantation, which was counterbalanced by the occurrence of transplant-related mortality (TRM) [39,40,41]. The presence of a matched sibling donor as a HSCT indication has been replaced, in most contemporary protocols, by risk assessments based upon disease characteristics and response-related factors. Current HSCT indications in childhood AML are summarized in Table 2. Considering recent improvements in chemotherapy and the potential risk of acute and late toxicities after HSCT, the current practice restricts the use of HSCT in CR1 only to those AML patients with high-risk (HR) features. However, there is no universal agreement on the definition of HR disease and different criteria have been, and continue to be, used by different cooperative groups. There is general consensus that standard-risk patients should not be transplanted in CR1 but only after the first relapse and achievement of a second complete remission. In this regard, the underlying genetic and molecular aberrations represent a major criterion for risk group stratification and allocation to HSCT. In particular, genomic alterations involving core-binding factor (CBF) transcription factors, namely inversion16(p13;1q22), t(16;16)(p13;q22) and t(8;21)(q22;q22), are widely recognized by all study groups as favorable risk group markers. Recently, a further chromosomal aberration involving CBF, t(16;21)(q24;q22), resulting in RUNX1-CBFA2T3 fusion, has been shown to be associated with good prognosis [42]. Noteworthy, a different translocation involving chromosomes 16 and 21, t(16;21)(p11;q22) (FUS-ERG), identifies instead a rare subgroup of AML with extremely poor prognosis, which should be considered for HSCT in CR1 [42]. In patients harboring t(8;21)RUNX1-RUNX1T1, an MRD reduction of less than 2 Log with respect to diagnosis, is associated with a significant higher risk of relapse when compared to patients who reduce MRD more than 3 Log. Although these subjects should be considered candidates for more aggressive therapies, additional studies are needed to determine whether they could benefit from an allogeneic HSCT in CR1 [43]. Less frequent genetic abnormalities, such as biallelic mutations of CEBPα, mutations in nucleophosmin1 (NPM1) with a normal karyotype, are also associated with a favorable prognosis in pediatric AML [44,45]. Patients with acute promyelocytic leukemia (APL) and t(15;17), owing to the advent of ATRA and arsenic trioxide, currently represent a group with an extremely favorable prognosis [46].

In the pioneering work from the Medical Research Council (MRC)-AML group, adverse cytogenetic features were originally defined as −5, −7, del(5q), abnormal 3q or complex karyotype [47]. Additional MRC studies focused on childhood AML identified abnormalities of 12p as a new cytogenetic group associated with poor prognosis [48]. Although these alterations cumulatively account for <5% of cases of childhood AML, several groups have demonstrated that their identification correlates with high rates of induction failure and poor survival, and there is general agreement that these patients should be offered allogeneic HSCT in CR1 [3,49,50,51,52]. The t(6;9)(p22;q34), which leads to the formation of a leukemia-associated fusion protein DEK-NUP214, occurs infrequently in children (less than 1% of AML cases) and is associated with FLT3 ITD in approximately 40% of cases. AML patients harboring t(6;9) have a high risk of treatment failure, particularly those not proceeding to allogeneic HSCT [48,53,54]. FLT3 internal tandem duplication (FLT3/ITD) occurs in approximately 10% to 20% of pediatric AML cases and conveys a poor prognosis, which is favorably modified by the presence of a low allelic ratio or concomitant NPM1 mutations. By contrast, patients harboring FLT3-ITD with a high allelic ratio and without an NMP1 mutation have a very high risk of relapse and benefit from HSCT in CR1 [55,56,57,58,59]. KMT2A (MLL) rearrangements (which occur in 20% to 24% of all patients with childhood AML) have a different prognostic value depending on the specific fusion partner. Results of a large, retrospective analysis including data from 756 pediatric patients with KMT2A-rearranged AML, showed that the t(4;11)(q21;q23.3)/KMT2A-MLLT2, t(6;11)(q27;q23)/KMT2A-MLLT4, t(10;11)(p12;q23)/KMT2A-AF10 and t(10;11)(p11.2;q23)/KMT2A-ABI1 were associated with a dismal outcome [60]. Patients with such abnormalities are almost unanimously considered candidates for allo-HSCT in CR1 [48,50,61,62]. Cryptic gene fusions, including NUP98-rearrangements (the most common being t(5;11)(q35;p15)/NUP98-NSD1 and t(11;12)(p15;p13)/NUP98-KDM5A) [63,64,65], CBFA2T3-GLIS2 (resulting from a cryptic inversion of chromosome 16) [65,66,67] and t(7;12)(q36;p13)/MNX1-ETV6 (occurring virtually exclusively in children younger than 2 years) [50,68] have been shown to predict poor outcome and warrant consolidation with allogeneic HSCT in CR1 [66,68,69,70]. Finally, the t(8;16)(p11;p13) rearrangement, fusing KAT6A to the CREBBP gene, has an age-dependent impact on prognosis. While in very young children this translocation is associated with spontaneous remission encouraging a watch-and-wait strategy, in older children the prognosis is poor and consolidation with allotransplant in CR1 should be seriously considered [71]. 

MRD measurement by multiparametric flow cytometry (MFC-MRD) has been recently shown as a strong and independent prognostic marker of relapse in pediatric AML. In a prospective study including 232 children with AML in which MFC-MRD was adopted as risk-stratification criteria together with genetic features, high-level MRD positivity (≥1% leukemic cells) after first induction was associated with a greater cumulative incidence of relapse MRD compared to low-level (<1%) MRD (49.2% ± 7.4% vs. 16.7% ± 7.8%, *p* < 0.0001). While the outcome for patients with low levels of MRD after first induction was identical to that of patients with negative MRD, detectable MRD levels below 1% after second induction were associated with a poor outcome [5]. Subsequent studies performed by different cooperative groups have confirmed the strong prognostic relevance of MRD after the first and the second induction course, and MFC-MRD is being increasingly employed to refine the current strategies of disease risk stratification, including identification of candidates for HSCT in CR1 [72,73,74,75].

Regardless of the initial risk classification, in relapsed AML allogeneic HSCT offers the best chance of cure, ideally after the achievement of second CR (CR2) [2,62,76]. In subjects who relapse after a previous HSCT, a second transplant can offer remarkable long-term disease-free survival (DFS) probability, even in those cases not obtaining a further remission [77,78,79,80]. The proportion of children with refractory AML, defined as failure to achieve a morphological remission after two courses of chemotherapy, is estimated to be as high as 10% [3,81]. Allogeneic HSCT is currently considered as the only curative strategy in these subjects, being capable of producing long-term DFS in up to 50% of cases [82]. Patients below the age of 10 years and those with low leukemia burden or in CR at the time transplant have the highest chance of cure [82,83]. 

## 4. The Choice of Conditioning Regimen

Since maximal reduction of leukemia cells is of highest importance for post-HSCT outcome in childhood AL, myeloablative conditioning (MAC) is still considered the gold standard [84]. 

For what concerns patients affected by ALL, total body irradiation (TBI)-containing regimens are used preferentially over chemotherapy-based ones. This was based on: (i) historical data [85]; (ii) retrospective/registry studies [84,86,87,88,89]; (iii) a small randomized controlled trial [90]. Hyperfractionated TBI, combined with cyclophosphamide (CY), was firstly used by investigators at Memorial Sloan Kettering Cancer Center [85]. A number of studies analyzed the effect of different chemotherapy agents in combination with TBI. Thiotepa (TT) in combination with CY was shown to be safe and effective [91]. In a CIBMTR study, the use of high-dose etoposide in combination with TBI compared favorably with TBI and CY for children with ALL in CR2 [86]. The same conditioning regimen was employed in the ALL-SCT-BFM-2003 study with good efficacy and manageable toxicity, resulting into low incidence of TRM [92]. For this reason, this association was chosen for the prospective randomized FORUM trial (see below). Interestingly, in another retrospective study, melphalan was identified as the best single agent in association with TBI, because of lower relapse incidence as compared with other drug combinations [93]. In view of notable well-known late side effects related to the use of TBI (including growth impairment, gonadal dysfunction, cognitive dysfunction, cataracts and secondary malignancies) [94], an important question is if TBI-based regimens can be replaced by chemotherapy-based conditioning. The combination of TBI and CY resulted into improved OS and leukemia-free survival (LFS) as compared to busulfan (Bu) and CY in a retrospective study on 627 patients [87]. A small, randomized, controlled trial in 43 children with ALL showed higher event-free survival (EFS) when a TBI-based regimen was used as compared to chemotherapy alone [90]. Interestingly, in a recent retrospective registry study on more than 3000 patients conducted by the Pediatric Disease Working Party of EBMT, although TBI was confirmed to be superior to chemotherapy in the whole cohort (for OS, LFS, TRM and relapse), a subgroup analysis showed comparable outcomes of the different type of conditioning regimens for patients in CR1 [84]. In order to obtain a definitive answer on the optimal preparative regimen to be employed in children with ALL, an international, prospective, open-label, randomized, controlled trial to investigate whether chemo-conditioning regimens could replace TBI in pediatric patients with high-risk ALL (For Omitting Radiation Under Majority age (FORUM), NCT01949129) was conducted in 21 countries in patients aged 4–21 years at HSCT, in CR pre-HSCT, and with an HLA-compatible related or unrelated donor [95]. In detail, fractionated 12 Gy TBI combined with etoposide was compared to fludarabine, TT, and either Bu or treosulfan (according to the country preference chosen by national coordinators of the study). The statistical design was a noninferiority study with an 8% margin, with an estimated sample size of 1000 patients randomly assigned in 5 years with 2-year minimum follow-up. However, the trial was terminated early because of application of a stopping rule, since at interim analysis chemo-conditioning resulted significantly inferior to TBI. In particular, 2-year OS (91% versus 75%, *p* < 0.0001) and EFS (86% versus 58%, *p* < 0.0001) were significantly higher for the 212 patients randomly assigned to receive TBI as compared to 201 who were randomized to chemotherapy alone (intention-to-treat analysis). Cumulative incidence of relapse and, more surprisingly, TRM were lower in the TBI group than in the chemotherapy-based preparative regimen. The outcome of children given fludarabine, TT and either Bu or treosulfan was comparable. Notably, in subgroup analyses, TBI remained superior to chemotherapy for almost all the different variables analyzed (including disease status) [95]. Thus, this kind of conditioning is now considered the gold standard for pediatric patients affected by ALL. 

In the context of acute myeloid leukemia (AML), TBI and non-TBI regimens have never been compared in a prospective, randomized fashion. The superiority of TBI-based compared to Bu-based regimens, initially reported in older studies present in literature [96,97,98] has not been confirmed in other reports. A retrospective study from the Japanese Society for Hematopoietic Cell Transplantation comparing TBI/Cy and intravenous Bu/Cy in pediatric AML patients in first or second remission (CR1/CR2) confirmed no significant differences in terms of relapse and TRM, this resulting in a similar 3-year OS (68% and 72% for TBI/Cy and Bu/Cy, respectively) [99]. Similar results have also been reported from the French Group in a retrospective study evaluating TBI/Cy versus Bu/Cy for AML patients in CR1 [100]. Recently, TBI and non-TBI regimens were compared in 624 children transplanted between 2008 and 2016 and reported to CIBMTR registry. Five-year NRM was higher with TBI regimens (22% vs. 11%, *p* < 0.0001), but relapse was lower (23% vs. 37%, *p* < 0.0001) compared to non-TBI conditioning. Consequently, OS (62% vs. 60%, *p* = 1.00) and LFS (55% vs. 52%, *p* = 0.42) did not differ between treatment groups. TBI regimens were associated with higher incidence of grade II–IV aGVHD and greater 3-year incidence of gonadal failure or growth hormone deficiency while no differences in cGVHD and late pulmonary, cardiac or renal impairment were observed [101].

The first association of Bu and high-dose cyclophosphamide (Cy) (200 mg/kg), reported from Santos et al., resulted into a low relapse rate, but unacceptable transplant related mortality (TRM): for this reason, the Cy dosage was reduced to 120 mg/kg in the subsequent studies, with a lower toxicity and the same efficacy profile [88]. The addition of a third alkylating agent (Melphalan) to a standard Bu/Cy backbone has been shown to be effective with an acceptable toxicity profile, and this conditioning regimen has been adopted by many cooperative groups [61,102]. Results from the Italian protocol AIEOP AML 2001/02, in which 243 patients given either allo- or auto-HSCT after myeloablative regimen including Bu/Cy/Mel were analyzed, showed a 8-year OS and DFS of 75% and 74%, the with cumulative incidence of TRM being of 7% for the allo-HSCT group [61]. Similar results were observed in the AML SCT-BFM 2007 trial [62]. A recent large retrospective study on behalf of the Pediatric Disease Working Party of the EBMT compared the outcomes of pediatric AML patients receiving three different conditioning regimens (TBI/Cy, BU/Cy, BU/Cy and melphalan). Among the three options, Bu/Cy/Mel was associated with a lower incidence of relapse and the higher LFS as compared with the other chemo/radiotherapy combinations [103].

In most recent years, the use of reduced intensity conditioning (RIC) has been explored in pediatric patients with AL as a strategy to maintain a good balance between high anti-leukemic effect and reduced acute/late chemotherapy toxicity. In this regard, the reduced-intensity approach could offer better outcome and minor toxicity to pediatric patients with AML who are ineligible for myeloablative therapy because of heavy pretreatment, comorbidities or genetic conditions at high-risk of transplant-related toxicities, including Fanconi anemia or Down syndrome [104,105]. Recently, in a large cohort of 180 AML patients, Bitan et al. reported no significant differences in of acute and chronic GvHD and 5-year OS and FFS between patients given or RIC and MAC regimens [106]. Treosulfan-based conditioning regimen appears an interesting alternative to busulfan [107,108]. EBMT data regarding 198 pediatric patients with hematological malignancies including 50 AML conditioned with Treosulfan-based regimens showed an optimal safety profile; the 3-year OS and EFS were 54% and 45%, respectively, with better results if Treosulfan was associated with fludarabine and an alkylating drug such as Tiothepa or Melphalan [108]. In a prospective II phase study evaluating safety and efficacy of a conditioning regimen based on Treosulfan, Fludarabine and Thiotepa in 29 pediatric patients with AML, the 3-year PFS and OS were 79% and 84%, respectively [109]. Results on other RIC or reduced toxicity regimens have been published in smaller cohorts of pediatric patients, reporting promising safety and efficacy results [110,111,112,113]. Of note, use of RIC or reduced toxicity regimen was explored in children with relapsed/refractory (R/R) AML in association with Clofarabine. Six patients with R/R AML underwent a reinduction with Clofarabine or high-dose Cytarabine and subsequently received RIC regimen and HSCT; at 21 months after the procedure, 5/6 patients were alive without relapse [114]. More recently, a large study including 103 CR1 AML pediatric patients compared safety and efficacy of Bu/Cy, Bu/Cy/Mel and Clofarabine/Flu/Bu; 5-year LFS was 43%, 59% and 67%, respectively, while 5-year TRM was 14%, 7% and 6%, respectively [115].

Anti-human T-lymphocyte globulin (ATLG) is largely used during the preparation to an allograft from donors other than an HLA-identical sibling to regulate bidirectional alloreactivity; indeed, this type of serotherapy reduces the incidence and severity of the two main immune-mediated complications of the procedure, namely GvHD and graft rejection. Several randomized trials conducted in adults have demonstrated a significant reduction in the incidence of chronic GvHD when ATLG is added to the standard GvHD prophylaxis [116]. Only one randomized controlled trial, comparing different doses of ATLG, has been conducted in children with hematological malignancies transplanted from an unrelated volunteer [117]. This study demonstrated that low-dose ATLG can reduce the incidence of life-threatening infections, without significantly affecting the incidence of acute and chronic GvHD; in addition, a lower dose of ATLG resulted into an improved OS and EFS. Thus, a low-dose ATLG (namely 15 mg/kg of Grafalon^®^, Neovii Biotech, Rapperswil, Switzerland; formerly ATG Fresenius^®^), should be regarded as the standard serotherapy regimen for HSCT from UD in pediatric patients with malignant disorders.

## 5. The Role of Pre-Transplant Minimal Residual Disease: Better Remission for Better HSCT Outcome?

During the last two decades, MRD quantification has emerged as a crucial assessment in the evaluation of early treatment response and in the definition of patient risk stratification in both ALL and AML [5,16,118,119,120]. At the same time, accumulating evidence has shown that pre-transplant MRD status correlates with the risk of relapse and OS after HSCT. In the late 1990s, Knechtli et al. provided the first demonstration of the key predictive role exerted by pre-transplant PCR-based MRD in a retrospective analysis of 64 pediatric patients undergoing allogeneic HSCT [121]. This observation was confirmed prospectively by the ALL-REZ BFM Study Group in 91 children with relapsed ALL. In this study, children with pre-HSCT PCR-MRD ≥10^−4^ had a higher cumulative incidence of relapse as compared with patients having PCR-MRD <10^−4^ (57% vs. 13% respectively, *p* < 0.001) [122]. The best cutoff value of pre-transplant PCR-MRD for prognosis prediction in ALL is still a matter of debate. In a large cohort of high-risk relapsed ALL children transplanted in CR2, subjects with PCR-MRD ≥10^−3^ before HSCT had a significantly worse probability of DFS as compared with patients with PCR-MRD <10^−3^, while no differences were observed between patients with MRD <10^−4^ and those having an MRD within the range of 10^−4^ and 10^−3^ [123]. By contrast, a retrospective analysis of 119 ALL patients, performed by the Italian Association for Paediatric Haematology/Oncology (AIEOP), showed that the level of pre-HSCT MRD positivity has a different impact on EFS according to disease phase at HSCT. Indeed, in patients transplanted in CR1, only an high PCR-MRD level (≥10^−3^) was associated with an increased risk of relapse, while in subjects transplanted in CR2, even a low-level MRD positivity (<10^−3^) determined a high relapse rate and poor outcome [124]. In the context of childhood AML, the prognostic value of molecular MRD before HSCT is less defined. In a recent I-BFM-AML collaborative study, the role of PCR-MRD collected within 5 weeks prior to HSCT was evaluated in 108 pediatric AML patients harboring one of the main recurrent AML gene rearrangements (t(8;21)(q22;q22); RUNX1-RUNX1T1, inv(16)(p13.1q22)/t(16;16)(p13.1;q22); CBFB-MYH11, t(9;11)(p22;q23); KMT2A-MLLT3 or FLT3-ITD). In this study, 5-year OS after HSCT was significantly higher in patients with low PCR-MRD (defined as a value below 2.1 × 10^−4^ calculated by ROC curve analysis with respect to diagnosis or relapse), as compared with subjects having PCR-MRD levels above the cutoff (83% vs. 57%; *p* = 0.012) [125].

The assessment of MRD by means of MFC (MFC-MRD) showed similar results in terms of prognostic value in both ALL and AML. In a cohort of 122 children with very high-risk ALL (*n* = 64) or AML (*N* = 58), Leung and colleagues showed that the 5-year cumulative incidence of relapse after HCT was 40% in the patients with high levels of FCM-MRD (≥0.1% in ALL and ≥1.0% in AML), 16% among those with low level of MRD (0.01% to <0.1% in ALL and 0.1% to <1% in AML) and 6% in those with no MRD (*p* = 0.0002). High MRD was also confirmed as an independent adverse prognostic factor for survival in multivariate analysis (*p* = 0.0035) [126]. In a Children’s Oncology Group (COG)/Pediatric Bone Marrow Transplant Consortium (PBMTC) multicenter phase III trial evaluating the addition of sirolimus to standard GvHD prophylaxis in children with ALL, patients with MFC-MRD ≥0.1% had a higher relapse risk as compared to subjects whose MRD was negative or <0.1% [127]. In the context of AML, detectable MFC-MRD immediately prior to HSCT has been consistently associated with increased risk of post-HSCT relapse and worse OS in both children and adults [10,128,129]. These observations are further supported by a meta-analysis of 19 studies evaluating pre-HSCT MRD (mainly assessed by FCM) in 1431 pediatric and adult AML patients; MRD positivity was associated with decreased DFS, OS and increased cumulative incidence of relapse [130]. Several groups have also shown the value of next-generation sequencing (NGS) technologies for MRD detection in both ALL and AML [131,132,133,134]. When NGS-MRD was compared with MFC-MRD in 56 pediatric B-ALL patients, NGS-MRD predicted relapse and survival more accurately than MFC-MRD (*p* < 0.0001), especially in the MRD negative cohort (2-year relapse probability, 0% vs. 16%; *p* = 0.02; 2-year OS, 96% vs. 77%; *p* = 0.003) [132]. Given the importance of pre-HSCT MRD in determining the probability of cure, modern ALL and AML therapy approaches have focused on strategies to induce MRD-negativity before transplant, especially for patients with high-risk disease features. These approaches are particularly attractive in B-cell ALL (B-ALL) where children may benefit from the tremendous improvements obtained over the past decade owing to the clinical application of different immunotherapy agents. Indeed, before the advent of immunotherapy, strategies to achieve pre-transplant MRD-negative remission in relapsed B-ALL relied on intensive multidrug chemotherapy regimens, which are commonly associated with the occurrence of toxicities that may be fatal or reduce the likelihood of proceeding to allogeneic HSCT [135,136].

In a phase I/II study conducted in children with R/R B-ALL, blinatumomab, a bispecific T-cell engager antibody targeting CD19, was able to induce cytomorphological remission in 39% of subjects, which was MRD-negative in 52% of cases [137]. Higher CR rates were observed in the expanded access study, likely because of the greater proportion of patients with <50% blasts enrolled, these data reflecting the association between lower leukemia burden and clinical response [138]. Recently, the results of two international randomized clinical trials, comparing blinatumomab with conventional chemotherapy as pre-transplant consolidation therapy in children with high-risk first relapse of B-ALL, were published. Randomization was performed after three cycles of chemotherapy in one study and after a single cycle of induction chemotherapy in the other one. In both studies, treatment with blinatumomab resulted in less severe toxicities, higher MRD remission rate, greater probability of proceeding to allogeneic HSCT and improved outcome [139,140]. 

In the dose-finding part of a phase 1/2 study promoted by the Innovative Therapies for Children With Cancer in Europe (ITCC) consortium, inotuzumab ozogamicin (InO), an antibody–drug conjugate composed of a CD22-directed monoclonal antibody linked to calicheamicin, was able induce a CR/CRi (CR with incomplete hematologic recovery) in 80% of children with relapsed and refractory B-ALL. Among the 19 responders with available MFC-MRD data, 16 (84%) obtained MRD-negativity [141]. Similar results were observed in a retrospective analysis of InO compassionate-use program, which reported a CR rate of 67% with 71% of responders achieving MRD negativity [142]. One possible toxicity concern, regarding the use of InO before HSCT, is related to the risk of sinusoidal obstruction syndrome (SOS), especially in heavily pretreated patients [141,142]. While blinatumomab and InO are mainly recognized as bridge-to-transplant (or, at least, bridge-to-consolidation) strategies in children, the benefit of a consolidative HSCT after chimeric antigen receptor T-cell (CAR T) therapy is currently the object of considerable debate [13]. Anti-CD19 CAR T-cell therapy has produced impressive MRD-negative CR rates, ranging from 56% to 93%. However, durability of response after CAR T is variable and influenced by CAR T-cell expansion, persistence and characteristics of co-stimulatory domains [11,143,144]. Studies with CD28-based anti-CD19 CAR T cells, which are characterized by short persistence after infusion, have a higher tendency of referral to HSCT compared to studies with 41BB-based CAR T cells. In the long-term follow-up study of 50 children and young adults (CAYA) treated with a CD28-based anti-CD19 CAR T, 62.0% patients achieved a CR, which was FCM-MRD negative in 90.3% of cases. Noteworthy, MRD-negative patients proceeding to allo-HSCT had a 5-year EFS of 61.9%, while all MRD-negative subjects who did not proceed to a consolidative HSCT experienced leukemia relapse [143]. In the global phase II trial of tisagenlecleucel in 75 CAYA with R/R B-ALL, only eight patients among those who obtained disease remission underwent allo-HSCT. Despite that, 59% of patients who received tisagenlecleucel remained in remission; the majority of those who relapsed experiencing CD19-negative disease [11]. A benefit from consolidative HSCT after anti-CD19 CAR T cell therapy has also been observed with a different 4-1BB-co-stimulated CAR [144],particularly in patients with rapid loss of B-cell aplasia and those who were not transplanted before CAR-T cells [145]. Consolidation with HSCT was shown to be favorably associated with better EFS (*p* = 0.016) also in a phase I trial evaluating a CD22-targeted/4-1BB CAR T-cell in CAYA with B-ALL [146]. Emerging data suggest that more sensitive NGS-MRD testing may help identify which patients need HSCT consolidation after CAR-T cell therapy [147]. In the near future, the dramatic increase in the number of B-ALL patients who achieve MRD-negative complete remissions owing to immunotherapy approaches, will likely offer the opportunity to re-evaluate the role of HSCT as consolidation strategy in this setting. 

For patients with AML and T-cell ALL, the availability of immunotherapy is currently much more limited as compared with B-ALL. In the context of pediatric AML, modest benefit has been observed by the incorporation of CD33-targeted therapy with gemtuzumab ozogamicin (GO) in addition to standard chemotherapy [148]. Despite that, there is evidence that certain subsets of patients, particularly those with FLT3-ITD mutations, KMT2A rearrangements, single-nucleotide polymorphisms in ABCB1 and CD33, and high CD33 expression are more likely to profit from GO administration and GO may be effective at reducing MRD levels before HSCT [5,149,150,151,152,153]. Concerns regarding the risk of increased toxicity of GO in conjunction with HSCT warrant further investigation regarding optimal dosing and timing to improve overall outcomes [151]. Several immunotherapies are in various stages of preclinical and clinical development for AML, including antibody–drug conjugates, bispecific antibodies, cellular therapies and checkpoint inhibitors [154,155]. Currently, most early-phase cellular immunotherapy studies for children with AML are intended as a bridge to transplant in order to achieve more profound remission status. It is too early to speculate whether such approaches will also be able decrease the need for subsequent HSCT [154,155]. Daratumumab, a CD38-targeting monoclonal antibody, is the most promising antibody-based approach in T-ALL treatment [156,157]. Currently, daratumumab in addition to standard chemotherapy is under investigation in a phase II trial for pediatric and young adult patients with R/R T- or B-cell ALL (NCT03384654). Developing CAR-T cell therapies into the setting of T-ALL has been hampered by the risk of fratricide because of the shared expression of target antigens between CAR-T cells and T-leukemia cells, and of severe life-threatening immunodeficiency due to the elimination of normal T lymphocytes [158]. Fratricide-resistant CD7, CD5 and CD1a-targeted CAR-T [159,160,161], and universal allotolerant off-the-shelf CAR-T cells generated by genome editing [162,163] have been proposed as potential strategies to overcome these limitations. 

Although accumulating evidence shows a correlation between the presence of pre-HSCT MRD and the risk or relapse in children with leukemia, merely having a detectable disease prior to HCT does not necessarily indicate an inability to cure the disease [126]. For this reason, especially in settings such as T-ALL and AML where the availability of immunotherapy approaches outside clinical trials is still limited, the benefit of repeated efforts aimed at achieving MRD-negativity before transplant should be carefully weighed against the risks of inducing additional toxicities affecting post-transplant outcome.

## 6. Post-Transplant Minimal Residual Disease: Is There Room for Intervention?

MRD assessment before transplantation cannot effectively identify all individuals with impending post-transplantation relapse who might benefit for pre-emptive intervention. For this reason, the predictive role of post-transplant MRD has been investigated by several groups, especially in ALL [124,164,165,166]. In a BFM study evaluating 113 pediatric patients transplanted for relapsed ALL, the level of PCR-MRD was inversely correlated with EFS and positively correlated with the cumulative incidence of relapse at all time points after transplant. In multivariable analysis, MRD ≥ 10^−4^ leukemic cells was consistently correlated with inferior EFS [164]. Although high levels of post-transplant MRD are strongly predictive of disease recurrence, low level MRD positivity after transplantation was not invariably associated with relapse, especially if detected early after HSCT [164]. By contrast, this and other studies showed that the further the patient is from HSCT, the more likely even low levels of MRD predict a poor prognosis [124,164,165,166]. These findings support the assumption that low levels of residual leukemia cells could be controlled by an immunologic GvL effect in the early post-transplant period, before the graft becomes tolerant toward the recipient. In a recent multicenter study, Bader et al. defined the relative risk of pre- and post-HSCT MRD in pediatric ALL, and their relationship with other independent risk factors [9]. When considered individually, however, both pre-HSCT and post-HSCT had significant prognostic value, if the two measures were simultaneously evaluated in a bivariate analysis, pre-HSCT MRD became less important in determining risk compared with the post-HSCT MRD. In this study, both MRD negative or positive patients had an approximately threefold decrease in relapse if they developed aGvHD and patients who had positive MRD recorded post-HSCT and developed aGvHD had relapse rates similar to those who were MRD negative and did not develop aGvHD [9]. The beneficial effect exerted by the occurrence of aGvHD on relapse risk and survival of children with ALL has been documented by several reports. In a COG/PBMTC study, patients with pre-HCT MFC-MRD ≥0.1% who did not develop aGvHD compared with those with MFC-MRD <0.1% who did develop aGvHD had much worse 2-year DFS (18% vs. 71%; *p* = 0.001). Patients with pre-HCT MRD <0.1% who did not experience aGvHD had higher rates of relapse than those who did develop aGvHD (40% vs. 13%; *p* = 0.008) [166]. Zecca et al. showed that children with hematologic malignancies who developed chronic GvHD (cGvHD) after transplant had a reduced relapse probability (16% ± 3% vs. 39% ± 3%, *p* = 0.0001) and a better DFS (68% ± 4% vs. 54% ± 3%, *p* = 0.01) compared with children without cGVHD, The anti-leukemic effect of cGvHD was observed mainly in patients with ALL [167]. In patients with B-ALL, post-HSCT NGS-MRD positivity performed better than FCM-MRD for predicting relapse, especially early after HCT. Any post-HSCT NGS positivity resulted in an increase in relapse risk by multivariate analysis (hazard ratio, 7.7; *p* = 0.05) [132]. The role of MRD after transplant in pediatric AML is less defined, although there is evidence that post-HCT positivity of MFC-MRD can predict imminent relapse [129]. Noteworthy, a recent study evaluated the predictive role of NGS-MRD in pediatric and adult AML patients undergoing HSCT. In this study, variant allele frequency (VAF) more than 0.2% on day +21 post-HSCT, was associated with decreased 3-year OS and an increased risk of relapse [133]. Finally, although less sensitive than MRD, close chimerism monitoring on peripheral blood has proven useful for the early detection of impending relapse in both ALL and AML [168,169,170].

For patients with post-transplant evidence of MRD, additional interventions could influence outcomes by favouring the development of a GvL effect. This goal has been pursued through several strategies, including rapid discontinuation (or abrupt cessation) of immune suppression [168,171,172,173], administration of cytokines [174] and infusion of donor-derived lymphocytes or cytokine-stimulated immune effector cells [170,175]. Since the benefit derived from GvL may be offset by the increased TRM associated with severe GvHD, caution should be adopted when adopting interventions that stimulate excessive GvHD [9]. In patients with Philadelphia chromosome positive ALL, or FLT3/ITD-positive AML, post-transplant administration of tyrosine kinase inhibitors, such as imatinib and sorafenib or midostaurin, respectively, can improve outcomes, especially for patients showing molecular MRD recurrence after-HSCT [176,177,178]. In the context of AML, the combination of azacytidine and donor-lymphocyte infusion (DLI) could represent an effective strategy to prevent overt disease recurrence in MRD-positive patients [179]. Recently, Ofran et al. described three patients with T-ALL in whom residual MRD after transplant was eradicated following administration of daratumumab, provided on a compassionate basis, in combination with DLI or rapid tapering of immunosuppression [157]. For patients with B-cell ALL, the availability of highly effective immunotherapy agents, such as blinatumomab and inotuzumab, also makes them attractive strategies to tackle MRD-recurrence in the post-transplant period. Ongoing clinical trials are evaluating these approaches in both children and adults (NCT04785547, NCT04044560, NCT03913559, NCT03104491). 

## 7. Transplantation from Alternative Donors: No Longer a Second Choice?

Eligibility criteria for HSCT in both ALL and AML have been varying over time not only according to disease characteristics and response to treatment, but also depending on type of available donor. The use of HLA-matched related donors is still considered the preferred option [40,92,180,181]. However, more than 70% of children with AL who might benefit from an allograft lack an HLA-identical family donor. With the establishment of donor registries, many patients are able to locate a suitable unrelated donor (UD). Results in children transplanted for either ALL or AML suggest that fully matched UD selected through high-resolution typing of HLA loci offer minimal or possibly no disadvantages in terms of disease outcome compared with HLA-identical siblings [182,183,184,185,186,187]. In the context of ALL, it has been shown that results obtained using a 9/10 or a 10/10 allelic matched donor, either related or unrelated, are not inferior to those observed after HSCT from an HLA-identical sibling in terms of EFS, OS and CIR [92]. For several years, a wider degree of HLA mismatch has been considered acceptable only in the presence of poor prognostic features indicating an increased risk of leukemia recurrence/progression without HSCT consolidation [92,188], although recent results obtained with different alternative donor approaches strongly suggest that this paradigm could no longer be valid. 

UCB transplantation (UCBT) has been largely employed in the past for patients lacking an HLA-matched donor and several reports have demonstrated that unrelated UCBT is able to offer long-term outcomes similar to those observed using an UD in both ALL and AML [189,190]. Eapen et al. compared results observed in 503 children with AL given unrelated UCBT with those of 282 bone marrow transplant (BMT) recipients. DFS was superior in recipients of HLA matched cord blood (*p* = 0.040), while it did not differ between BMT and one or two HLA-disparate UCBTs [190]. The results of UCBT in both childhood ALL and AML have been extensively analyzed by retrospective registry-based studies performed by the Eurocord group. In the context of ALL, a lower risk of relapse (24% vs. 39%; hazard ratio 0.4; *p* = 0.01) and better DFS (54% vs. 29%; hazard ratio 2, *p* = 0.003) were observed in children with negative MRD before UCBT as compared with those with positive MRD [191]. In children given UCBT for AML, the results were particularly promising for children with poor prognostic features, namely secondary leukaemia, high-risk karyotype, or transplanted in CR2 after an early relapse, for whom the DFS and risk of relapse were similar to those of the other patients [192]. Analysis of AML patients who underwent autologous or allogeneic HSCT within the AIEOP AML 2002/01 showed a remarkable DFS overcoming 90% in the subgroup who received UCBT [61]. Subsequent studies have demonstrated that better HLA-matching strategies between donor and recipient (at high-resolution level for HLA-A, -B, -C and -DRB1 loci) may further improve the outcome of patients with AL given a single UCBT [193,194]. Transplantation of two UCB grafts has been proposed to overcome the limitations related to the low cell dose infused; unfortunately, in two prospective randomized studies, the double UCBT strategy did not improve the overall outcome of children and young adults with AL in the presence of a single unit of adequate cell dose, being instead associated with an increased risk of GvHD [195,196]. Despite that, it has been suggested that the double-unit strategy may enhance the GvL effect and be of particular benefit in patients with positive pre-transplant MRD [197,198]. Strategies aimed at expanding the number of UCB progenitors ex vivo and favouring stem-cell-homing in vivo are being developed with encouraging preliminary results, holding the potential to revitalize the field of UCBT in the near future [199,200,201].

One of the main reasons for the continue decline in the utilization of UCBT observed in the last decade, beyond the intrinsic limitations of the approach (i.e., delayed engraftment, increased risk of TRM), is the emergence of highly effective T-cell depleted and T-cell repleted HLA-haploidentical transplantation platforms [202]. Indeed, HSCT from an HLA-haploidentical relative offers an immediate transplant treatment virtually to any patient lacking a matched donor or a suitable UCB unit and, in addition, easy access to the donor for post-transplant adoptive cell therapies [203,204].

In the setting of T-cell depleted (TCD) haploidentical (haplo) HSCT, a remarkable advance has been obtained through the development of more sophisticated graft manipulation strategies (with respect to the original positive selection of CD34+ cells), based on the selective depletion of CD3+ and CD19+ cells [187,205,206] or, more recently, of TCRα/β+ and CD19+ cells [207]. With the latter approach, in particular, the graft transfers to the recipient not only high numbers of CD34+ cells, but also of mature donor NK cells and TCRγ/δ+ T cells that can display their protective effect against leukemia regrowth and life-threatening infections [208]. By contrast, T cells expressing the αβ chains of the T cell receptor, which are responsible for the development of GvHD, are removed with a median depletion efficiency of 4 Log [209]. Our group has reported the outcome of a cohort of 80 children with AL (ALL, *n* = 56; AML, *n* = 24) transplanted from an haploidentical relative after αβ+ T-cell and CD19+ B-cell depletion [210]. All children received a fully myeloablative preparative regimen and received anti-T-lymphocyte globulin for preventing graft rejection and GvHD; no patient received any post-transplant GVHD prophylaxis. The cumulative incidence of TRM was remarkably low (5%). Nineteen subjects relapsed, the cumulative incidence of recurrence being 24%. The 5-year OS probability was 72% (95% CI, 62–82) for the whole study population. Overall, the 5-year DFS was 71% (95% CI, 61–81), without differences between ALL and AML patients. The cumulative incidence of skin-only, grade I–II acute GvHD (aGvHD) was 30% and no patient developed GvHD with visceral involvement, grade III–IV aGvHD or extensive chronic GvHD. TBI-containing preparative regimen was the only variable favourably influencing relapse incidence and chronic GVHD-free/relapse-free survival (GRFS) [210]. More recently, the results of 98 αβ-TCD haplo-HSCT recipients were compared with those of 127 matched UD (MUD), 118 mismatched UD (MMUD) in a multicenter retrospective analysis. MUD and MMUD-HSCT were characterized by a higher cumulative incidence of grade II–IV and grade III–IV aGvHD (35% vs. 44% and 6% vs. 18%, respectively) compared with 16% and 0% in αβ-TCD haplo-HSCT (*p* < 0.001). Children treated with αβ-TCD haplo-HSCT also had a significantly lower incidence of overall and extensive chronic GVHD (*p* < 0.01). While the GRFS probability of survival of MUD-HSCT and αβhaplo-HSCT recipients was superimposable (61% and 58%, respectively), the choice of MMUD-HSCT had a detrimental effect on this composite end-point (34%; *p* < 0.001) [6]. Noteworthy, although the haploidentical donor was mainly selected privileging NK immunological features [211,212], no favourable influence of NK alloreactivity and of donor KIR B-haplotype was observed in both studies [6,210]. This is in contrast with the observations derived from studies on the infusion of CD34+ cells [213,214,215,216], likely because the NK-mediated GVL effect was partially obscured by other cells present in the graft, including γδ T cells [217]. In a retrospective Spanish study performed in children with ALL given ex vivo TCD grafts using different manipulation strategies, the presence of donor–recipient KIR mismatch provided no advantage, while a donor with KIR A haplotype was associated with an increased probability of relapse [218]. Although the benefits of an NK alloreactive asset appear less clear when using refined graft-manipulation strategies, the selection of a donor with favourable NK features continues to be recommended for patients receiving TCD-HSCT [212]. Since preclinical data showed that bisphosphonates can enhance TcRγδ-mediated anti-leukemia activity [219], the administration of zoledronic acid (ZOL) after αβ-TCD haplo-HSCT has been explored as potential strategy for further improving patient outcome, showing encouraging results in terms of better OS, reduced GvHD incidence and lower TRM [220,221]. In addition, TCD haplo-HSCT approaches represent an ideal platform for post-transplant adoptive cell therapies because no (or minimal) immunosuppression is given after HSCT. An intriguing approach to accelerate the recovery of adaptive immunity and to promote anti-tumour activity relies on the administration of suicide gene-modified T cells, which offer the possibility of limiting the risk of uncontrolled GvHD by triggering T-cell apoptosis [222]. In the adult TCD haplo-HSCT setting, post-transplantation infusion of donor T-cells modified with the insertion of the herpes simplex thymidine kinase suicide gene (to achieve in vivo susceptibility to ganciclovir) enabled regulation of GvHD, while promoting immune reconstitution [223,224]. An alternative strategy, developed by the Baylor group, is based on T cells engineered to express the caspase 9 (iC9) suicide gene [225]. Post-transplant infusion of iC9-T cells has been shown to accelerate immune recovery [226,227]. Administration of iC9-T cells after αβ+ TCD-haplo-HSCT was associated with better OS and DFS when compared with an historical cohort of “pure” αβ+ TCD-haplo-HSCT [228]. In order to promote the recovery of pathogen-specific immunity, other groups have investigated the infusion of low-dose memory CD45RA-depleted donor lymphocytes after HSCT with αβ T-cell depletion, with encouraging results [229]. The rationale for this strategy is based on experimental data demonstrating that mouse CD4 memory T-cells, and effector memory CD8 T-cells, are devoid of GvH reactivity [230]. Removal of CD45RA+ *naıve* T lymphocytes has been also tested as TCD strategy before haploidentical HSCT. In a recent publication describing the preliminary outcomes of 50 children receiving a CD45RA-depleted haplo-HSCT (of whom 47 having AL), the St. Jude investigators reported moderately high rates of grade III–IV aGvHD (28%) and chronic GvHD (26%). Despite that, GvHD was successfully treated in most patients, and NRM mortality among patients in CR at the time of HCST was 5.6%, the 3-year OS and EFS being 78.9% and 77.7%, respectively [231]. 

T-cell repleted haplo-HSCT approaches based on the post-transplant administration of cyclophosphamide (PTCy), pioneered by the Johns Hopkins group [232], have been extensively investigated in adult patients [233,234]. In the last few years, accumulating evidence suggests that this strategy may also be successfully employed in children. A recent retrospective analysis compared the outcome of pediatric patients with AL undergoing T-replete haplo-HSCT with PTCy after nonmyeloablative (NMA) conditioning with that of a contemporary cohort of children receiving transplantation from MUD and MMUD. The 5-year OS rates were not different between MUD, MMUD and Haplo patients, while an increased incidence of graft failure was observed in the haplo-group, likely derived from the adoption of an NMA preparative regimen [235]. A full-myeloablative regimen, either busulfan-based or TBI-based, has been employed with success in 29 children and young adults undergoing HLA-haplo-HSCT with PTCy for malignant conditions (of whom 18 with AL), with 3-year OS and EFS of 79% (95% CI, 66, 96) and 69% (95% CI, 54, 88), respectively. In this small cohort, relative high rates of cGvHD were observed (cumulative incidence of cGvHD at 1-year 28% (95% CI, 0.1, 0.4), moderate to severe 14% (95% CI, 0.01, 0.3)), but the limited sample size prevents drawing definitive conclusions [236]. The results of a retrospective analysis of 180 children with ALL who received a HLA-haplo-HSCT with PTCy, were recently reported by Pediatric Disease Working Party (PDWP) of the European Society for Blood and Marrow Transplantation (EBMT). Cumulative incidence of day-100 grade II–IV aGvHD was 28%, and 2-year cGvHD was 21.9%. The 2-year NRM was 19.6% and the estimated 2-year DFS was 65%, 44% and 18.8% for patients transplanted in CR1, CR2 and CR3, respectively. In multivariable analysis for patients in CR1 and CR2, disease status (CR2 (hazard ratio 2.19; *p* = 0.04)), age at HCST greater than 13 years (hazard ratio 2.07; *p* = 0.03) and use of peripheral blood stem cell (PBSC) (hazard ratio 1.98; *p* = 0.04) were independently associated with decreased OS [237]. Recent evidence also suggests that refinements in pharmacological GvHD prophylaxis strategies may improve the outcome of patients receiving unmanipulated HLA-mismatched grafts from UD. In particular, in a prospective multicenter phase II trial, the addition of abatacept, an anti-CTLA-4 monoclonal antibody, to standard calcineurin inhibitor (CNI)/methotrexate (MTX)-based GvHD prophylaxis was able to reduce the incidence of grade III–IV aGvHD in children and adults with haematological malignancies undergoing HSCT from an 8/8 MUD or 7/8 MMUD. Patients receiving abatacept had better severe aGvHD-free survival (SGFS) as compared with controls receiving only CNI/MTX (93.2% vs. 82, *p* = 0.05 in the 8/8 arm; 97.7% vs. 58.7%, *p* < 0.001, in the 7/8 arm) [238]. The impressive results observed in the 7/8 arm are of particular interest for those patients (often belonging to ethnic groups poorly represented in the registries) for whom the possibility of UD-HSCT is mainly restricted to an HLA-mismatched setting [239]. 

Collectively, most recent transplant results obtained with MUD, MMUD, haploidentical donors and UCB units indicate that all these options are able to offer the chance of transplant to virtually every child with AL in need of an allograft and lacking an HLA-identical sibling, without significant differences in terms of outcome. Each of these strategies has advantages and limitations, but rather than being considered competing alternatives, they should be regarded as complementary options and the final choice should be based on patient’s characteristics, clinician/Centre expertise, the urgency of the transplant and specific features of the unrelated/haploidentical donor or UCB unit.

## 8. Conclusions

Results of HSCT in AL have improved substantially, in particular during the last two decades, and progress using alternative donors has extended the potential application of allotransplantation to most patients. For several years, donor characteristics have played a relevant role in determining eligibility for HSCT, with alternative donors being accepted only in those cases deemed at the highest risk of leukemia relapse. However, current results with alternative donors suggest that this paradigm could no longer be valid and encourage not restricting the indications for HSCT upon availability of a fully matched donor. In this regard, the importance of the specific transplant infrastructure and experience of the team in determining the outcome of HSCT from alternative donors must not be neglected [240]. The importance of MRD before and after transplant is being increasingly clarified and MRD-directed interventions may further improve the outcome of allogeneic HSCT in AL, particularly in view of the growing availability of precision medicine approaches. In addition, incorporation of MRD data in risk stratification models before allo-HSCT may allow better identifying which patients should be offered an allograft and possibly inform how to transplant these subjects [241]. In the last few years, successes obtained by combination chemotherapy and the development of highly effective immunotherapy agents have raised the question of whether HSCT will continue to play a role in modern therapy of childhood AL, particularly in B-ALL [13]. Although the answer to this question should necessarily come from the results of well-designed and randomized clinical trials, the number of variables that may influence transplant outcome and the quantity of innovative approaches that have entered, or are about to enter, into the clinical arena, makes the design and conduction of such studies anything but trivial. For these reasons, determining the appropriate role of HSCT in childhood AL will continue to be a challenging and dynamic process, requiring constant and careful assessment of the likelihood of cure with conventional chemotherapy or novel targeted therapies to identify the subset of children for whom transplant offers a better treatment option. In this regard, close cooperation between chemotherapy cooperative study groups and transplant/cell therapy societies, and monitoring of treatment-related late effects with long-term patient follow-up is crucial.

## Figures and Tables

**Table 1 jcm-10-03790-t001:** HSCT indications in pediatric patients with ALL (adapted from AIEOP-BFM ALL 2017 and IntReALL 2010 protocol).

**CR1**	TCF3-HLF, irrespective of MRD results	
	Positive MRD at TP1 or TP2 (irrespective of MRD value) and: - No CR at d33 - KMT2A-AFF1 - hypodiploidy <44 chr. or DNA index <0.8 - IKZF1plus and intermediate-/high-risk MRD results - T-ALL + poor-prednisone response and/or MFC-MRD d15 >10%	
	MRD TP2 ≥5 × 10^−4^	
	Ph+ positive ALL and MRD ≥5 × 10^−4^ at end Induction IB	
	Infants (age < 1 year) with KMT2A-rearrangments and: - age < 6 months and initial WBC > 300,000/μL - age < 6 months and Prednisone Poor-Response - No CR at d33 - MRD TP2 ≥5 × 10^−4^	
**CR2**	HR	Very early isolated extramedullary relapse of BCP or T-ALL
		Any bone marrow relapse of T-ALL (irrespectively of the time elapsing between diagnosis and relapse)
		Very early BCP-ALL bone marrow relapse
		Early BCP-ALL bone marrow relapse
		Very early BCP-ALL combined bone marrow relapse
	SR	Late isolated or combined bone marrow relapse of BCP-ALL and poor MRD response at the end of induction therapy *
		Early combined bone marrow relapse and poor MRD response at the end of induction therapy *
		Early isolated extramedullary relapse of BCP or T-ALL
**>** **CR3**	All patients	

TCF3-HLF: TCF3-HLF positive acute lymphoblastic leukemia; MRD: minimal residual disease; CR: complete remission; MFC-MRD: flow-citometry MRD; WBC: white blood cells; BCP: B-cell precursor ALL (acute lymphoblastic leukemia). T-ALL: T-cell ALL; TP1: time point 1; TP2: time point 2. Time point of relapse: Very early, <18 months after primary diagnosis; early, ≥18 months after primary diagnosis and <6 after completion of primary; late, ≥6 months after completion of primary therapy. SR: standard risk; HR: high risk. * cut-off is defined by the specific treatment arm.

**Table 2 jcm-10-03790-t002:** HSCT indications in childhood AML.

**CR1**	**GENETIC RISK CRITERIA**
	Complex karyotype (≥3 aberrations including at least one structural aberration)
	Monosomal karyotype (−7, −5, del 5q)
	11q23/KMT2A rearrangements, involving: - t(10;11)(p12;q23)/*KMT2A-AF10* - t(10;11)(p11.2;q23)/*KMT2A-ABI1* - t(6;11)(q27;q23)/*KMT2A-MLLT4* - t(4;11)(q21;q23.3)/*KMT2A-MLLT2*
	t(11;12)(p15;p13)/*NUP98-KDM5A*
	t(7;11)(p15.4;p15)/*NUP98-HOXA9*
	t(5;11)(q35;p15)/*NUP98-NSD1*
	t(6;9)(p23;q34)/*DEK-NUP214*
	t(16;21)(q24;q22)/*RUNX1-CBFA2T3*
	t(7;12)(q36;p13)/*MNX1-ETV6*
	t(3;21)(26.2;q22)/*RUNX1-MECOM*
	t(16;21)(p11.2;q22.2)/*FUS-ERG*
	*FLT3*-ITD with AR ≥0.5 without NPM1 mutations
	inv(3)(q21.3q26.2)/t(3;3)(q21.3q26.2)/*RPN1-MECOM*
	inv(16)(p13.3q24.3)/*CBFA2T3-GLIS2*
	12p abnormalities
	**RESPONSE RISK CRITERIA**
	MRD ≥ 1% after the first induction course
	MRD ≥ 0.1% after the second induction course
	Primary Induction Failure [i.e. patients with ≥25% blasts after the first induction course and ≥5% blasts after the second induction course]
	**SECONDARY AML**
	Therapy-related AML
	AML evolving from myelodysplastic syndrome (MDS)
**≥CR2**	All patients

HSCT: hematopoietic stem cell transplantation; NPM1: nucleophosmin1.

## Data Availability

Not Applicable.

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
