# Peer review of "The Role of Allogeneic Hematopoietic Stem Cell Transplantation in Pediatric Leukemia"

_jcm, 2021, doi:10.3390/jcm10173790_

Round 1
Reviewer 1 Report
I liked this extensive review by Algeri et al. The authors discuss important clinical points.
I have few comments:
It is great that the authors discuss in detail about prior studies and existing evidence. However, at times you get lost following all these data. I would suggest to make all the data reporting concise and focus on the aim of the section and the point you are trying to make. The results from some studies can be reported as combined and do not need specific details all the time.
Minor comments:
Just for the sake of format, I would suggest to discuss HSCT indications in ALL similar to how it has been done in AML. I would suggest to take away bullets and discuss in paragraph instead.
Overall, english language is good. But I would suggest shortening sentences so that it is easy to follow, especially with all the data being discussed.
Author Response
REVIEWER 1
Comments and Suggestions for Authors
I liked this extensive review by Algeri et al. The authors discuss important clinical points.
We would like to thank the Reviewer for appreciating our manuscript.
I have few comments:
It is great that the authors discuss in detail about prior studies and existing evidence. However, at times you get lost following all these data. I would suggest to make all the data reporting concise and focus on the aim of the section and the point you are trying to make. The results from some studies can be reported as combined and do not need specific details all the time.
As recommended by the Reviewer, we have shortened the text, eliminating some specific details and focusing on the main messages contained in the papers that we discussed in this review article.
Minor comments:
Just for the sake of format, I would suggest to discuss HSCT indications in ALL similar to how it has been done in AML. I would suggest to take away bullets and discuss in paragraph instead.
As suggested by the Reviewer, we have deleted the bullets.
Overall, english language is good. But I would suggest shortening sentences so that it is easy to follow, especially with all the data being discussed.
As recommended by the Reviewer, whenever possible, we have tried to shorten the length of the sentences.
Reviewer 2 Report
An exciting, carefully prepared publication summarizes what is know about HSCT in pediatric patients. A crucial topic for pediatric hematologists. There are no significant criticisms of the text. Still, without tables and figures summarizing the subject of the publication, it won't be easy to understand. Additionally, the literature emphasizes the role of iron metabolism disorders in children undergoing HSCT (https://doi.org/10.3390/cancers13123029). The authors themselves should briefly discuss this topic.
Author Response
An exciting, carefully prepared publication summarizes what is know about HSCT in pediatric patients. A crucial topic for pediatric hematologists. There are no significant criticisms of the text.
We would like to sincerely thank the Reviewer for the positive global evaluation of our manuscript.
Still, without tables and figures summarizing the subject of the publication, it won't be easy to understand.
As recommended by the Reviewer, we have included 2 tables that summarize the indications for HSCT in childhood ALL and AML.
Additionally, the literature emphasizes the role of iron metabolism disorders in children undergoing HSCT (https://doi.org/10.3390/cancers13123029). The authors themselves should briefly discuss this topic.
We respectfully think that this topic is not strictly pertinent to the scope of this review article. In particular, the paper cited by the Reviewer is difficult to integrate in the text.
Round 2
Reviewer 2 Report
The authors have addressed all the comments of the reviewers and revised the manuscript accordingly.